# Impact of the Persistence of Three Essential Oils with Antifungal Activities on Stored Wheat Grains, Flour, and Baked Products

**DOI:** 10.3390/foods10020213

**Published:** 2021-01-21

**Authors:** Zaida N. Juárez, Horacio Bach, María E. Bárcenas-Pozos, Luis R. Hernández

**Affiliations:** 1Chemistry Area, Deanship of Biological Sciences, Universidad Popular Autónoma del Estado de Puebla, 21 Sur #1103 Barrio Santiago, Puebla C.P. 72410, Mexico; 2Department of Medicine, Division of Infectious Disease, University of British Columbia, 2660 Oak Street, Vancouver, BC V6H 3Z6, Canada; hbach@mail.ubc.ca; 3Department of Chemical and Food Engineering, Universidad de las Américas Puebla, Ex Hacienda Sta. Catarina Mártir S/N, San Andrés Cholula 72810, Mexico; maria.barcenas@udlap.mx; 4Department of Chemical Biological Sciences, Universidad de las Américas Puebla, Ex Hacienda Sta. Catarina Mártir S/N, San Andrés Cholula 72810, Mexico; luisr.hernandez@udlap.mx

**Keywords:** *Porophyllum linaria*, *Agastache mexicana*, *Buddleja perfoliata*, essential oils, flour, cookies, persistence interval

## Abstract

Wheat grains are exposed to several plagues after harvesting and during storage. These plagues include bacteria, fungi, and insects with detrimental outcomes to their quality and heavy losses to the farmers. Fungi are of special interest because of their ability to produce mycotoxins with health concerns. Once grains are harvested, synthetic fungicides, which are sprayed before long-term storage, normally control fungi; however, these synthetic products represent a health concern because of their toxicities. Previously, we reported the antifungal activity of the essential oils extracted from *Porophyllum linaria*, *Agastache mexicana*, and *Buddleja perfoliata* against fungal strains isolated from stored wheat. In this study, we sprayed wheat grains with the same essential oils to measure their persistence interval and to prepare baked products to assess potential changes in their physical properties. The persistence interval of the essential oils in grains indicated that it takes between 63 and 134 days to eliminate 90% of the original compounds. This extended time of the compounds in the grains together with a lack of physical properties modifications of the flour and baked products (post-treatment) suggest that the presence of oils in the grains is potentially safe to use. The solid data denote the technological feasibility of the treatment and the possible management of residues through adequate safety intervals.

## 1. Introduction

Wheat is one of the first cereals domesticated by humans and is now one of the most important mainstays of food for humanity, constituting the main source of carbohydrates [1].

Grains are harvested and stored with a variety of organisms, including bacteria, fungi, and insects. These organisms represent a potential concern if grains are not processed properly before their storage. One example is a fast deterioration of their quality, affecting not only the nutritional value but also generating odors and color changes which may affect their milling and baking qualities. 

Many fungal strains have been found on wheat grains, including the genera *Alternaria*, *Cladosporium*, *Drechslera*, *Epicoccum*, *Eurotium*, *Fusarium*, *and*
*Penicillium* [2,3]. Besides the economic loss caused by fungi growing in stored grains, some of these species are mycotoxin producers. These toxins are dangerous to human health because they are responsible for many pathologies, among them cancer and allergies [4,5]. 

Fungi are controlled in stored grains with synthetic fungicides that in many cases have an unknown action mode with toxic effects on humans and animals, causing a negative effect on the environment [6]. It is desirable to search for harmless alternatives with minimal or no toxicity to humans. One option is the use of essential oils (EOs), which have been reported as an effective means to control fungal growth [7,8,9,10,11]. 

Recently, we reported the antifungal activity of essential oils extracted from the plants *Porophyllum linaria*, *Agastache mexicana* ssp. *xolocotziana*, and *Buddleja perfoliata* [10,11]. These essential oils showed significant activity against fungal strains isolated from stored wheat grains with no significant cytotoxic effects on mammalian cells tested in an ex vivo model using macrophages derived from human monocytes (THP-1 cells) [10,11]. To further advance the applications of these oils, we added them to wheat grains and after about two months, we proceeded to obtain the flour from these treated grains and elaborate dough to assess its quality for baking. The persistence interval of each essential oil was also determined to see their persistence in wheat grains and to predict their behavior over time through appropriated mathematical models.

## 2. Materials and Methods

### 2.1. Plant Material and Wheat Grains

*Porophyllum linaria* (Cav.) (voucher 0049), and *Agastache mexicana* ssp. *xolocotziana* Bye, E. Linares & Ramamoorthy (voucher 0120) were collected on 21 May 2014 at the Ethnobotanical Garden Francisco Peláez Roldán, placed in San Andrés Cholula, Puebla, Mexico. The biologist Estela Hernández Ascención identified the plants, and a specimen of each plant was deposited in the same garden under the voucher numbers indicated above in parenthesis. *Buddleja perfoliata* (Kunth) was bought in the Puebla market in January 2012 and classified in the herbarium of the Benemérita Universidad Autónoma de Puebla by the botanist Allen J. Coombes. A specimen of this plant was deposited in this herbarium with voucher number HJ-075/2012.

The wheat grains used in this study were organic *Triticum aestivum* harvested in “Rancho Los Álamos” (Tlaxcala, Mexico), having an organic certificate number 2014-090 issued by MAYACERT and according to the National Organic Program 7 CFR Part 205 of USDA.

### 2.2. Extraction and Physical and Chemical Characterizations of the Essential Oils

The extraction and chemical characterization of the essential oils used in this study was previously described [10,11].

The density (ρ) of the essential oils was calculated by weighing 1 mL of the oil using an analytical scale. Experiments were performed in triplicate. 

The refractive index (RI) was measured using a VEE GEE refractometer (Abbe Refractometer-C10, Vernon Hills, IL, USA) at 26 °C. The refractometer was calibrated using distilled water (RI = 1.334) and 1-bromonaphtalene (Cargill) (RI = 1.656) and according to the method 58-20.02 of the AACCI [12]. All measurements were performed in triplicate.

The color of the essential oils was determined by a portable Konica Minolta colorimeter (CR-400, New York, NY, USA) following the method 58-12.01 [12]. The transmittance mode was used for the two liquid essential oils (*A. mexicana* and *B. perfoliata*) using the following color parameters: L * (luminosity, white-black), a * (green-red), and b * (blue-yellow) in the Hunter scale. The colorimeter was calibrated using the black mosaic and the calibration parameters: L * = 100, a * = 0, and b * = 0. The measurements were performed in a quartz cell. In the case of the semisolid oil from *P. linaria* as well as the flours and cookies detailed below, the same parameters were measured but using the CIELab scale. The colorimeter was calibrated using the white standard mosaic (L * = 94.43, a * = −1.03 y b * = +0.80). Colors of the essential oils, flours, and cookies were measured in triplicate.

### 2.3. Persistence Interval of Essential Oils

In this work, the persistence interval of essential oils is defined as the time that all or some of the initial constituents of the essential oil remain in the wheat grain. The persistence interval was determined as follows: 4 g of wheat grains were placed in a glass vial, filling only one-third of its volume. On the other hand, an acetone solution of the corresponding oil was prepared in enough amount to obtain a final concentration equal to the highest minimal inhibitory concentration (MIC, µg of oil/g of wheat) as previously reported [10,11]. Subsequently, with the help of a micropipette, the wheat grains were impregnated with the appropriate amount of solution homogeneously distributed among the grains. The bottles were covered in a non-hermetic manner with aluminum foil and stored in a chamber with controlled temperature and relative humidity at 25 °C and 50%, respectively, with a photoperiod of 12 h of light and 12 h of darkness. The wheat grain impregnated with difenoconazole dissolved in acetone (30 µg/mL) was used as a positive control, while the negative controls were the wheat grains alone (untreated) and the wheat grains impregnated with acetone. Triplicate tests were performed and analyzed weekly by gas chromatography coupled to mass spectrometry (GC-MS). For the chromatographic analysis, a Varian-CP3800 gas chromatograph (San Diego, CA, USA) coupled to a Varian-4000 mass detector (San Diego, CA, USA), in Head Space mode and a VF-5 ms, 30 m long, 0.25 mm in diameter and 0.25 µm thick, capillary column was used, with helium as carrier gas (1.0 mL/min). The temperature ramp in the column was started at 60 °C for 2 min, increased to 280 °C at a rate of 10 °C/min, maintaining the temperature for 6 min. A volume of 1 mL of the oil vapors was injected into the column in splitless mode, using a Pal System autosampler (Combi Pal RF546 model). The autosampler was programmed with GC Head Space injection mode using a 1 mL syringe; the temperature of the syringe was 60 °C and that of the agitator was 180 °C, with incubation of 3 min and a stirring cycle of 2 s at 500 rpm and waiting for 4 s. The injector temperature was set at 250 °C. The mass spectrometer was operated at 70 eV, and the mass range was 40 to 425 uma. The persistence interval of each of the essential oils was evaluated by comparing the sum of the area of all peaks corresponding to the essential oil on day zero (beginning of the experiment) and that corresponding on the day of control. The persistence interval was evaluated for 7 weeks.

### 2.4. Flour Preparation

Wheat grains (3.5 kg) were manually cleaned, placed in a 6 L beaker, and mixed with a solution of essential oils dissolved in acetone. Final concentrations of 30 μg/mL were used for *A. mexicana*, whereas 0.69 and 50 μg/mL were used for *P. linaria* and *B. perfoliata*, respectively reported [10,11]. These concentrations represent the reported minimal inhibitory concentrations (MICs) against the same fungal strains and are expressed as µg of essential oil/g of grain in this study. The solution of the essential oil was homogenously distributed using a micropipette. After 1 h of the essential oil application, the beakers were capped with a piece of aluminum foil, sealed, and stored in a chamber with controlled temperature and moisture of 25 °C and 50%, respectively. A solution of the fungicide difenoconazole (30 μg/mL in acetone) was used as the positive control, while untreated grains and grains impregnated with acetone were used as negative controls. All grain samples were stored for a period of 8 weeks after which the grains were conditioned for milling by increasing their humidity until reaching 16%. The milling was carried out at the laboratories of MEJORPLUS Company (Puebla, Mexico). Experiments were performed in triplicate.

Flour yields were calculated taking into consideration the relationship between the weights of the grains used for the milling and the weight of the flour obtained. Results were expressed in percentages.

### 2.5. Assessment of the Flour Qualities

The following parameters were measured in flour according to the AOAC International methods 925.10, 923.03, 920.86, 920.85, and 960.52 [12]: moisture, ashes, fiber, fat, and protein, respectively. For the blank, the obtained values in percentage were: 11.19 ± 0.002, 0.13 ± 0.07, 0.39 ± 0.37, 1.27 ± 0.34, and 8.21 ± 1.70, respectively.

The rheological properties of the flour were determined by farinograms obtained using a Brabender farinograph (300 g capacity, Duisburg, Germany) and performed according to the AACC International 54-21 method [12]. Alveograms were obtained using a Groupe Tripette and Renaud alveograph (Chopin, France) and according to AACC International 54-30 method [12]. All these tests were performed at MEJORPLUS Company laboratories (Puebla, Mexico).

The moisture was measured according to the AACC International 44-15A method [12]. Measurements were performed in triplicate.

### 2.6. Quality Parameters of Cookies Elaborated with Treated Flours

#### 2.6.1. Cookie Preparation

The cookie dough was made according to the following recipe: wheat flour (150 g), brewer’s yeast (4 g), sugar (8 g), salt (1.5 g), butter (30 g), vanilla extract (8 drops), corn oil (1 tablespoon), and water (6 tablespoons). All ingredients were mixed thoroughly, and the formed dough was manually extended and cut with a round mold. Cookies were baked in an oven at 220 °C for 18 min and cooled at room temperature.

#### 2.6.2. Cookie Characteristics

The water activity of the cookies was determined with an electric hygrometer (Aqualab 4 TEV, Pullman, Washington, DC, USA), and the measurements were performed in triplicate at 25 °C. The hardness was measured according to the AACCI-74-09.01 method [13] using a Shimadzu EZ-SX instrument (Kyoto, Japan) with a compress assay at 5 mm/s and a cylindrical probe. Each assay was repeated 30 times.

#### 2.6.3. Sensory Analysis

Sensory analysis was carried out with a triangular test based on the preparation of two samples using the same flour derived from the essential oil-treated grains and the third was made with flour without essential oil addition. The analysis was performed by responding to a survey with a panel of 30 referees [14].

### 2.7. Statistical Analysis

Data were analyzed using the Statgraphics Centurion XVI, ver. 16.2.04 program. A *p* value ≤ 0.05 was considered statistically significant.

In the case of persistence interval, a simple regression analysis was applied to the oil quantity data (measured as an area of the chromatogram) based on the elapsed days from the moment of the impregnation of the wheat grain, which is considered the zero-day. The data were tried to fit 27 different curvilinear models, choosing the one with the highest R^2^. In this way, the mathematical model that best predicts the persistence interval of each of the essential oils was obtained.

## 3. Results and Discussion

### 3.1. Physical Characterization of the Essential Oils

The refractive index, the density, and the color of the essential oils are reported in Table 1. The color of the essential oils was determined, and the essential oils of *A. mexicana* and *B. perfoliata* showed yellow tones, while the color of *P. linaria* was brown.

### 3.2. Persistence Interval of Essential Oils

#### 3.2.1. *Agastache mexicana* Essential Oil

Analysis of the persistence interval of the constituents in the grains impregnated with *A. mexicana* essential oil (EO) showed a decrease of approximately 30% during the first 10 days and then remained relatively stable over the next 20 days. After this period, a new decrease was measured after which it produced a new fall, to reach a loss of 79.2% at the end of the experiment (day 49) (Table 2).

The Log-Y Sq-X model was used to fit the observed behavior, where “Y” is the amount of the essential oil, represented by the sum of all GC peak areas; and “X” is the number of days of the experiment. With an R^2^ = 0.9098 and RMSE of 2.17 × 10^6^, the equation that represents the model is
EO amount as GC peak areas = exp(16.9806 − 0.000558591 * Day #^2).

#### 3.2.2. *Buddleja perfoliata* Essential Oil

The behavior of the *B. perfoliata* essential oil is quite different compared with that of *A. mexicana*. The essential oil amounts suffered a sharp fall to reach a loss of 66.9% during the first 14 days, and then, with a small slope reached a loss of 84.1% on day 49 (Table 3).

The best mathematical model that explains this behavior is Log-Y SqRt-X, with the same meaning for “Y” and “X” as above. With an R^2^ = 0.8933 and RMSE of 5.99 × 10^5^, the equation that represents the model is
EO amount as GC peak areas = exp(16.1116 − 0.244455 * sqrt(Day #))

#### 3.2.3. *Porophyllum linaria* Essential Oil

Due to the small value of the MIC calculated for this essential oil (which was used to impregnate the wheat samples), its persistence interval could only be evaluated until day 28. After this day, the persistence interval significantly dropped below the detection level of the equipment.

The behavior of this essential oil shows that the loss is lesser than those observed in the previously described essential oils with a constant but slow decay through time (Table 4). The best mathematical model that explains this behavior is Sq-Y SqRt-X, with the same meaning for “Y” and “X” as above. With an R^2^ = 0.6405 and RMSE of 1.14 × 10^4^, the equation that represents the model is
EO amount as GC peak areas = sqrt(6.23172E11 − 5.33363E10 * sqrt(Day #))

In conclusion, the behaviors of the three essential oils are different. Applying each equation, a loss of 90% of the persistence interval was measured after 63, 79, and 134 days for the essential oils of *A. mexicana*, *B. perfoliata*, and *P. linaria*, respectively.

### 3.3. Quality Parameters of Flours Obtained from the Treated Wheat Grains

#### 3.3.1. Yields

Flour yields were approximately 47% for all samples. The yield of the flours derived from essential oil-treated grains was 46%, while flours considered as blank, positive, and negative controls yielded 47, 46, and 47%, respectively. These results indicate that essential oils had no effects on flour yields. These yields were low compared to the normal yield obtained in industrial flours, which varies between 72–78% [15], suggesting that industrial milling is more efficient than laboratory milling.

#### 3.3.2. Moisture and Color

Moisture content is of critical importance in food because it affects its physical properties, including taste, texture, appearance, and shelf life. Moisture and color of the flours obtained from both treated and untreated (negative control) grains are shown in Table 5. Results showed that the moisture of flours treated with essential oils showed no significant differences compared to the negative control within an accepted range of <14% [16]. Similarly, the color was not significantly altered by the essential oils, except the L* parameter of the flour obtained from grains treated with *B. perfoliata* essential oil, which gave more brilliance to the flour.

#### 3.3.3. Rheological Properties

Evaluation of the rheological properties of the dough is an important factor that evaluates the flour quality because it studies the physical properties of the hydrated gluten that is formed during kneading [17]. The behavior of dough during kneading can be predicted from the farinograms. Typical bread flour presents stability of 10 min and a developing time of 3 min, while the content of water ranges between 55–60% [18,19]. To evaluate the dough workability, an alveogram was obtained. Typical bread flour presents strength between 120–160 × 10^4^ J, with an equilibrium (P/L) ranging between 0.4–0.6 [20,21]. In our study, the flours obtained from both treated and untreated grains showed similar farinograms and alveograms, suggesting that the essential oils had no significant effects on the rheological properties of the dough (Table 6 and Table 7).

### 3.4. Cookies Characterization

#### 3.4.1. Quality Parameters

Cookies were prepared with flours obtained from both treated and untreated wheat grains. Water activity, moisture, hardness, and color were measured and shown in Table 8.

The water activity, hardness, and moisture values showed significant differences when compared to the untreated flour, which can contribute to longer shelf life for cookies. On the other hand, the color parameters presented in Table 1 showed differences only in cookies made with flour obtained from grains treated with *P. linaria* essential oil. These differences may be due to the procedures, baking time, and tray location in the oven.

#### 3.4.2. Sensory Evaluation

The triangular test was the method chosen for this evaluation, using the minimal number of correct judgments table to establish significant differences [22]. According to this table, for a panel of 30 referees, at least 15 of them must identify differences among the cookies to conclude that there is a significant difference. Results showed that the number of judgments could not reach the number 15, suggesting that there were no significant differences between cookies made with flour obtained from grains treated vs. untreated grains.

### 3.5. Essential Oil Chemical Compositions and a Probable Interpretation of Dough Behavior

The wheat grain is composed of several layers. The most external ones are known as bran, and they are mainly constituted by fiber, but they also contain other compounds like lignins, oligosaccharides, phytic acids, polyphenols, phenolic acids, and other trace compounds [23]. On the other hand, as we previously reported [10,11], the *A. mexicana* and *B. perfoliata* essential oil constituents are mainly mono- and sesquiterpenes, while fatty acids and the diterpene phytol are the main compounds in *P. linaria* essential oil. Taking into consideration the nature of these compounds, it is expected that the essential oils will remain on the bran layer because of compound similarities. These interactions are likely intermolecular forces like hydrogen bonds or dipole interactions. Thus, the adsorption of the essential oils on the bran layer protects the whole grain against the fungi with limited penetration of the compounds towards the internal layers with limited effects on the flour quality.

## 4. Conclusions

Here, we present data related to the effect of *A. mexicana*, *B. perfoliata*, and *P. linaria* essential oils on the behavior of flours, showing no significant effect on the dough quality. In the case of cookies, there were no significant differences compared to the untreated flours regarding water activity, moisture, hardness, or color parameters, as well as no significant differences found in the sensory evaluation. In addition, we provide evidence that 90% of the constituents of the essential oils are lost in 63–134 days post-treatment, and according to the type of the essential oil.

Although the treatment reduced the growth of the fungal strains (probably by a slow release of the essential oil components), no assessment of the flour or baked products were performed. To the best of our knowledge, this is the first report of baking cookies using flour obtained from wheat grain treated with essential oils with antifungal activity, supporting the use of these essential oils as an alternative to the use of synthetic fungicides.

## Figures and Tables

**Table 1 foods-10-00213-t001:** Refractive index, density, and color parameters of essential oils.

Plant	Refractive Index	Density (g mL^−1^)	Color Parameters		
			L *	a *	b *
*A. mexicana*	1.5215	0.955 ± 0.001	84.1 ^a^ ± 0.3	1.8 ^a^ ± 0.2	4.5 ^b^ ± 0.3
*B. perfoliata*	1.5005	0.919 ± 0.002	70.5 ^b^ ± 0.8	−1.6 ^a^ ± 0.4	1.8 ^a^ ± 0.5
*P. graveolens*	1.4555	0.744 ± 0.0008	68.5 ^c^ ± 0.4	−1.5 ^a^ ± 0.4	2.1 ^a^ ± 0.4

No significance (*p* ≤ 0.05) is labeled with the same letter in the same column, ND: Not determined, * Measured in reflectance mode.

**Table 2 foods-10-00213-t002:** Persistence interval of *A. mexicana* essential oil.

Day	Mean Area *	Standard Deviation	Standard Error of the Mean	Percentage Standard Error of the Mean	% Decrease
0	2.65 × 10^7 a^	4.81 × 10^6^	2.77 × 10^6^	10.5	
14	1.92 × 10^7 b^	2.72 × 10^6^	1.57 × 10^6^	8.19	27.4
21	1.79 × 10^7 b^	1.23 × 10^6^	7.12 × 10^5^	3.97	32.2
28	1.78 × 10^7 b^	7.19 × 10^6^	4.15 × 10^5^	2.33	32.7
35	1.63 × 10^7 b^	9.03 × 10^5^	5.21 × 10^5^	3.19	38.3
42	9.45 × 10^6 c^	6.62 × 10^5^	3.82 × 10^5^	4.04	64.3
49	5.50 × 10^6 d^	1.41 × 10^5^	8.14 × 10^4^	1.48	79.2

* Essential oil amount calculated as the GC peak area. Values with the same letters (a, b, c or d) represent homogeneous groups after the Fischer test of least significant difference (LSD) at 95% confidence.

**Table 3 foods-10-00213-t003:** Persistence interval of *B. perfoliata* essential oil.

Day	Mean Area *	Standard Deviation	Standard Error of the Mean	Percentage Standard Error of the Mean	% Decrease
0	1.14 × 10^7 a^	2.37 × 10^6^	1.68 × 10^6^	14.8	
14	3.76 × 10^6 b^	3.75 × 10^5^	2.17 × 10^5^	5.77	66.9
21	2.72 × 10^6 b,c^	3.31 × 10^5^	1.91 × 10^5^	7.03	76.0
28	2.66 × 10^6 b,c^	5.25 × 10^5^	3.03 × 10^5^	1.14	76.6
35	2.42 × 10^6 b,c^	4.05 × 10^5^	2.34 × 10^5^	9.64	78.6
42	2.44 × 10^6 c^	2.37 × 10^5^	1.37 × 10^5^	5.61	78.6
49	1.80 × 10^6 c^	3.29 × 10^5^	1.90 × 10^5^	1.05	84.1

* Essential oil amount calculated as the GC peak area. Values with the same letters (a, b or c) represent homogeneous groups after the Fischer test of least significant difference (LSD) at 95% confidence.

**Table 4 foods-10-00213-t004:** Persistence interval of *P. linaria* essential oil.

Day	Mean Area *	Standard Deviation	Standard Error of the Mean	Percentage Standard Error of the Mean	% Decrease
0	7.89 × 10^5 a^	3.74 × 10^4^	2.16 × 10^4^	2.74	
14	6.52 × 10^5 a,b^	1.05 × 10^5^	6.05 × 10^4^	9.28	17.4
21	5.94 × 10^5 b^	9.38 × 10^4^	5.41 × 10^4^	9.12	24.7
28	5.91 × 10^5 b^	6.71 × 10^4^	3.87 × 10^4^	6.55	25.1

* Essential oil amount calculated as the GC peak area. Values with the same letters (a or b) represent homogeneous groups after the Fischer test of least significant difference (LSD) at 95% confidence.

**Table 5 foods-10-00213-t005:** Humidity and color parameters of flour from wheat treated with the essential oils.

Flour	Humidity (%)	Color Parameters		
		L *	a *	b *
Blank ^#^	11.190 ^a^ ± 0.002	96.17 ^a^ ± 0.04	0.03 ^a^ ± 0.04	8.98 ^a^ ± 0.13
*A. mexicana*	11.8800 ^a^ ± 0.0009	96.19 ^a^ ± 0.05	0.043 ^a^ ± 0.006	8.74 ^a^ ± 0.18
*B. perfoliata*	10.94 ^a^ ± 0.02	96.41 ^b^ ± 0.23	0.07 ^a^ ± 0.03	8.93 ^a^ ± 0.10
*P. linaria*	12.730 ^a^ ± 0.007	96.15 ^a^ ± 0.05	0.08 ^a^ ± 0.02	8.93 ^a^ ± 0.23

No significance (*p* ≤ 0.05) is labeled with the same letter (a or b) in the same column. ^#^ Flour from untreated wheat.

**Table 6 foods-10-00213-t006:** Alveogram characteristics of the flour obtained from wheat treated with essential oils.

	Untreated	Positive Control ^1^	Negative Control ^2^	*A. mexicana*	*B. perfoliata*	*P. linaria*
P (tenacity) (mm)	70.2	73.5	77.2	73.7	74.4	75.5
L (extensibility) (mm)	37.0	31.7	29.2	31.7	31.7	31.7
P/L (configuration of the curve)	1.90	2.31	2.64	2.32	2.35	2.38
W (baking strength) (×10^4^ J)	100	95.0	100	100	95.0	100

^1^ Wheat treated with acetone and difenoconazole (positive control). ^2^ Wheat treated with acetone.

**Table 7 foods-10-00213-t007:** Farinograph diagram parameters of the flour obtained from wheat treated with the essential oils.

	Untreated	Positive Control ^1^	Negative Control ^2^	*A. mexicana*	*B. perfoliata*	*P. linaria*
Water absorption (mL/100 g flour)	57.1	57.6	57.5	57.5	56.5	56.9
Arrival time (min)	0.50	0.75	0.50	0.75	0.50	0.75
Departure time (min)	2.00	1.75	1.75	2.00	2.00	2.50
Stability (min)	0.25	1.00	1.25	1.25	1.50	1.75
Peak time (min)	1.00	1.00	1.00	1.25	1.25	1.25
Mixing Tolerance Index (MTI) (Brabender Units)	100	120	120	120	110	110

^1^ Wheat treated with acetone and difenoconazole (positive control), ^2^ Wheat treated with acetone.

**Table 8 foods-10-00213-t008:** Water activity, humidity, hardness, and color parameters (L *, a *, b *) of cookies elaborated with flour from wheat treated with the essential oils.

Treatment	Water Activity	Humidity (%)	Hardness (N)	Color Parameters		
				L *	a *	b *
Untreated	0.44 ^a^ ± 0.01	4.88 ^a^ ± 0.002	42.14 ^a^ ± 10.39	85.02 ± 2.54 ^ab^	0.40 ± 0.18 ^ab^	20.70 ± 0.70 ^a^
*A. mexicana*	0.28 ^b^ ± 0.01	3.26 ^b^ ± 0.001	34.43 ^b^ ± 11.97	86.04 ± 1.27 ^b^	0.37 ± 0.10 ^a^	21.35 ± 0.48 ^b^
*B. perfoliata*	0.324 ^c^ ± 0.0006	3.13 ^b^ ± 0.0003	36.27 ^b^ ± 10.67	84.02 ± 1.36 ^a^	0.75 ± 0.21 ^c^	21.10 ± 0.70 ^ab^
*P. linaria*	0.346 ^d^ ± 0.006	4.48 ^c^ ± 0.002	35.23 ^b^ ± 12.37	85.61 ± 2.24 ^b^	0.49 ± 0.18 ^b^	19.70 ± 1.28 ^c^

No significance (*p* ≤ 0.05) is labeled with the same letter (a, b, c or d) in the same column.

## Data Availability

Data available on request.

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
