# Peer review of "Impact of the Persistence of Three Essential Oils with Antifungal Activities on Stored Wheat Grains, Flour, and Baked Products"

_foods, 2021, doi:10.3390/foods10020213_

Round 1

Reviewer 1 Report

The Manuscript is very interesting and represents an idea about a sustainable and widely available method to stop spoilage in wheat. However, there are some issues which need to be adressed :

- I recommend a title change: "Impact of residual levels of three essential oils with antifungal activities on stored wheat grains, flour and baked products"

- Page 8, lines 170 -171: you already mentioned moisture content determination on Page 7, line 161. Are two method deliberately used and why?

- Why were the residual levels of essential oils measured only in wheat grain, not in flour and cookies? I believe the residual levels of flour and cookies should also be reported, since the essential oils can have a big influence on sensory prooperties.

- Sensory evaluation based on there is a difference/there is no difference seems a little bit insufficient for a proper analysis. In my opinion, a more detailed sensory evaluation should have been conducted with tested properties ranging from textural sensations to odd taste, aftertaste etc. Or at least a backing analysis with E-tongue which cold define the differences between treated and untreated cookies should be included.

- Page 9, line 200: Please include RMSE values for the models. The R2 value often isn't good enough as a measure of fit if the RMSE values are too high.

- Data shown in Table 2 are also shown in Fig.1 and Fig.2. There is no need to show the same data in two ways, please leave either the table or the figures. The same goes for Table 3 and Figs 3 and for; and for Table 4 and Figs 5 and 6.

- Pages 19-20, lines 350-360. I do not see the point of this separate subsection. This text can easily be included in the introduction or the discussion section after the analysis of essential oils or before flour analysis.

- Page 20, line 369 : you state that the essential oils reduced the growth of the fungal strains. However, there is no data about the fungal growth monitoring in this manuscript. Furthermore, there is no description of a fungal growth monitoring method in the Materials and Methods section. Please supplement the manuscript with data on fungal growth.

Reviewer 2 Report

Parts of text to be corrected

39-42   “This extended time of the compounds in the grains together with a lack of physical properties modifications of the flour and baked products (post-treatment), suggest that the presence of oils in the grains are potentially safe to use.”

The safety judgment of the residues is based on data from previous, partial, insufficient experiences. The solid data denote the technological feasibility of the treatment and the possible management of residues through adequate safety intervals.

62        “carcinogenic and allergenic effects”

Mycotoxins are responsible for many pathologies, as a result of various pathogenetic mechanisms, not just those mentioned! The authors correct the consideration on the basis of scientific data, included those contained in the references reported.

72        “with no significant cytotoxic effects to mammalian cells”

Authors should correct by indicating the methodology by which cototoxicity was tested.

113-114          “In this work, the residual levels of essential oils were defined as the time that all or some of the initial constituents of the essential oil remain in the wheat grain.”

The definition of residual level should not involve the time factor. From a technical and legal point of view, the residue of a substance concerns the components of the substance and not the time.

Authors should correct the residual level meaning, possibly extrapolating it from technically and legally accepted terms. For example “the initial constituents of essential oil present in or on wheat grains or derived food products derived from.

Similarly, the time parameter could be derived from the concept of: safety interval or withdrawal period.

200-201          “In this way, the mathematical 200 model that best predicts the residual level of each of the essential oils was obtained.”

For the considerations made above, the residual term should be corrected with the term "safety interval".

Round 2

Reviewer 1 Report

The Authors have explained all of the questions raised in a satisfactory manner.